# Identification of Genes Related to Cold Tolerance and Novel Genetic Markers for Molecular Breeding in Taiwan Tilapia (*Oreochromis* spp.) via Transcriptome Analysis

**DOI:** 10.3390/ani11123538

**Published:** 2021-12-13

**Authors:** Pei-Yun Chu, Jia-Xian Li, Te-Hua Hsu, Hong-Yi Gong, Chung-Yen Lin, Jung-Hua Wang, Chang-Wen Huang

**Affiliations:** 1Department of Aquaculture, National Taiwan Ocean University, Keelung City 20224, Taiwan; kimi0396@gmail.com (P.-Y.C.); abcpeter92@gmail.com (J.-X.L.); realgigi@gmail.com (T.-H.H.); hygong@mail.ntou.edu.tw (H.-Y.G.); 2Center of Excellence for the Oceans, National Taiwan Ocean University, Keelung City 20224, Taiwan; 3Institute of Information Science, Academia Sinica, Taipei 11529, Taiwan; cylin@iis.sinica.edu.tw; 4Department of Electrical Engineering, National Taiwan Ocean University, Keelung City 20224, Taiwan; jhwang@mail.ntou.edu.tw; 5AI Research Center, National Taiwan Ocean University, Keelung City 20224, Taiwan

**Keywords:** Taiwan tilapia, cold-tolerance, RNAseq, microsatellite, single nucleotide polymorphism

## Abstract

**Simple Summary:**

In this study, we investigated the brain, gill, liver, and muscle transcriptomic responses of Taiwan tilapia towards cold stress. Some key genes and molecular markers involved in cold biological pathways were screened through differential expression. Among them, energy-related metabolic pathways and nucleotide genotypes were highly correlated with cold tolerance traits. This suggested that single nucleotide polymorphism (SNP) genetic variation can be used as a molecular marker to assist the selection and verification of cold-tolerant populations. Our study results will accelerate the understanding of different farmed tilapia tolerance mechanisms to environmental temperature changes and provide insights for the molecular breeding of cold-tolerant Taiwan tilapia species.

**Abstract:**

Taiwan tilapia is one of the primary species used in aquaculture practices in Taiwan. However, as a tropical fish, it is sensitive to cold temperatures that can lead to high mortality rates during winter months. Genetic and broodstock management strategies using marker-assisted selection and breeding are the best tools currently available to improve seed varieties for tilapia species. The purpose of this study was to develop molecular markers for cold stress-related genes using digital gene expression analysis of next-generation transcriptome sequencing in Taiwan tilapia (*Oreochromis* spp.). We constructed and sequenced cDNA libraries from the brain, gill, liver, and muscle tissues of cold-tolerance (CT) and cold-sensitivity (CS) strains. Approximately 35,214,833,100 nucleotides of raw sequencing reads were generated, and these were assembled into 128,147 unigenes possessing a total length of 185,382,926 bp and an average length of 1446 bp. A total of 25,844 unigenes were annotated using five protein databases and Venny analysis, and 38,377 simple sequence repeats (SSRs) and 65,527 single nucleotide polymorphisms (SNPs) were identified. Furthermore, from the 38-cold tolerance-related genes that were identified using differential gene expression analysis in the four tissues, 13 microsatellites and 37 single nucleotide polymorphism markers were identified. The results of the genotype analysis revealed that the selected markers could be used for population genetics. In addition to the diversity assessment, one of the SNP markers was determined to be significantly related to cold-tolerance traits and could be used as a molecular marker to assist in the selection and verification of cold-tolerant populations. The specific genetic markers explored in this study can be used for the identification of genetic polymorphisms and cold tolerance traits in Taiwan tilapia, and they can also be used to further explore the physiological and biochemical molecular regulation pathways of fish that are involved in their tolerance to environmental temperature stress.

## 1. Introduction

Due to the extreme climate and environmental changes caused by global warming that include unusual snowfall and frequent winter cold currents, all biological communities and ecosystems have been impacted [1]. The sustainable development of agricultural production stability and safety is a major challenge facing all countries today [2,3]. Taiwan is located in a tropical/subtropical region. The majority of aquaculture production is dominated by subtropical fish species. High temperatures in summer and typhoons, heavy rainfall, low temperature, and cold currents in winter along with other extreme abnormal weather have caused serious damage to aquaculture fisheries and economic losses [4].

Tilapiine fishes of the cichlid family are native to tropical and subtropical regions of Africa and eventually reached and settled in the Middle East through the Great Rift Valley [5]. Given the tropical origin of these fish, 20–30 °C is the optimal growth temperature for most tilapia. When the temperature is lower than 20 °C, feeding and reproduction are typically inhibited. Currently, in addition to many genetic studies examining low temperature tolerance in different tilapia species, a number of countries have also developed cultivated tilapia strains exhibiting different degrees of cold-tolerance through selection and hybridization [6,7]. In Taiwan, after years of research, genetic improvement, and certification, tilapia fish have gradually become a unique and high-quality strain. Twenty years ago, the industry termed these fish “Taiwan tilapia” [8].

Thermal tolerance is a quantitative trait that is biologically significant in several cultured fish species. This body phenotypic variation is attributed to acclimation, physiological stage, genetic effects, and environmental factors [9,10]. To survive, fish can also respond to environmental temperature fluctuations [11]. When fish are stimulated by the environment, they activate the two major physiological regulation nerves and endocrine systems in the body to maintain cellular homeostasis by regulating metabolism, osmotic pressure, and the endocrine system [12,13,14]. If the body functions of these fish are damaged, the resistance of these fish will decrease, and this can result in disease or even death.

At the molecular level, low temperatures reduce the enzymatic reaction rates and the diffusion and transport of biomolecules and can slow protein denaturation and disaggregation [15]. Additionally, low temperatures can also inhibit transcription and translation, destroy cellular cytoskeletal elements, alter membrane permeability, and affect energy production in cells [16]. Furthermore, previous genetic studies have used microsatellite markers to facilitate genetic linkage map quantitative trait loci (QTL) mapping [17] and cold-tolerance marker development [7,18,19]. Several studies have attempted to further explore the genetic model and the genetic basis of cold-tolerance in fish [20,21,22,23,24]. However, the mechanisms and pathways underlying the observed variations in cold-tolerance among tilapia fish species remain unknown.

In this study, genetic variation within species was characterized by selecting and breeding families possessing different low-temperature tolerances and sensitivities. Furthermore, transcriptome analysis was used to compare the transcriptome responses in brain, gill, liver, and muscle tissues between cold-tolerant and sensitive Taiwan tilapia strains under low-temperature challenges. In addition to characterizing the differential gene expression patterns between cold-tolerant and cold-sensitive fish, it is also possible to develop and verify the low-temperature test of the functional genes involved in the regulation pathway. Molecular genetic DNA markers for microsatellites and single nucleotide polymorphisms assist in broodstock breeding. In the future, this can accelerate the molecular selection of low-temperature-tolerant strains of Taiwan tilapia to mitigate the losses of the aquaculture fish industry due to cold damage and increase the economic value of the aquaculture industry.

## 2. Materials and Methods

### 2.1. Animals and Experimental Conditions

Three tested Taiwan tilapia strains were collected from the wild and from private markets by the National Taiwan Ocean University (NTOU), and breeding programs were performed at the core breeding greenhouse and Aquatic Organism Research and Conservation Center in Gongliao. The Rui-Fang (RF) strain is a wild species of Taiwan tilapia that was originally collected from the Rui-Fang field in northern Taiwan. The YiHua (YH) strain is a breeding species of Taiwan tilapia that was originally collected from the private commercial seed breeding farm in southern Taiwan. The NTOU (NT) strain is a pure strain of Nile tilapia, and this strain has been maintained for a long time period in the NTOU [25,26].

Spawns were obtained by mating in pairs to obtain 22 families (Appendix A). The offspring of each family were kept in separate fish tanks until they were 3 to 4 months old and weighed approximately 30–40 g or more. Subdermally injected dyes were used to identify families. To determine the cold-tolerance value of each family, each one contributed fry for cooling tests. The operation method was referred to and modified from the description published by Nitzan et al. [9].

### 2.2. Cooling Test

The selection criteria for cold-tolerance traits as assessed using the cooling test involved the use of F1 (*n* = 15) and F2 (*n* = 15) test fish from the offspring of full-sib and half-sib families that were raised in a 90-L orange bucket that was equipped with an upper filter and an air pump (TURBO 600 pumping motor, SanYu Aquarium, Kaohsiung, Taiwan). The upper portion was covered with a plastic fence to prevent the fish from jumping out of the tank. The fish were allowed to adapt to a water temperature of 25 °C for one week, and the photoperiod was set to 12 h light/12 h darkness. The upper portion of the filter was cleaned, the water was changed, and the fish were fed daily. The offspring fish of each family were then moved into a 600-L water tank that was equipped with a temperature-controlled system (Firstek Scientific, New Taipei City, Taiwan) that was used to slowly lower the water temperature at a rate of 1 °C per day. The water temperature was recorded and observed after each cooling. The swimming behavior and feeding behavior of Taiwan tilapia were observed, and a dissolved oxygen saturation of greater than 90% was maintained. Finally, the minimum tolerable temperature (°C), full length (cm), body length (cm) and mass (g) of each flipping fish was recorded, and tail fin tissue samples were collected to extract genomic DNA for subsequent analysis.

The offspring of each Taiwan tilapia family were tested for low-temperature tolerance using a cooling and cold attack test. The first three families (RF strain) considered most tolerant to low temperature, and the last three families (NT strain) considered least tolerant were selected and defined as the cold-tolerance (CT) and cold-sensitivity (CS) groups.

### 2.3. Total RNA Extraction

The cold-tolerance (CT, *n* = 6) and cold-sensitivity (CS, *n* = 6) fish groups were subjected to cold stimulation at 10 °C, and the brain, gill, liver, and muscle tissues were collected and placed into a 1.5 mL centrifuge tube for isolation (Roche Applied Science, Mannheim, Germany). The EasyPure Total RNA Spin Kit (Bioman, Taipei, Taiwan) was used for total RNA extraction. Three stainless steel beads (3 mm) and one (5 mm) steel ball (LabTurbo^®^, Taigen Bioscience, Taipei City, Taiwan) were placed into a 1.5 mL centrifuge tube containing Trizol, and they were homogenized in the SpeedMin PLUS (Analytik Jena AG, Biometra GmbH, Göttingen, Germany) and maintained at room temperature for 5 min. The mixture was poured into a 2 mL filter column and centrifuged (4 °C, 10,000× *g*) for 2 min. The supernatant was then transferred to a new centrifuge tube, and 400 µL of 70% ethanol was added. The tube contents were shaken, mixed immediately, and poured into the RB column (Bioman, Taipei, Taiwan). This was centrifuged (4 °C, 10,000× *g*) for 2 min, and the lower layer of liquid was discarded and the column was placed into a 2 mL collection tube. Next, 400 µL of W1 buffer was added to the RB column that was subsequently centrifuged (4 °C, 10,000× *g*) for 1 min. The lower layer of liquid was discarded and returned to the 2 mL collection tube. Then, 600 µL of wash buffer was added to the RB column that was subsequently centrifuged (4 °C, 10,000× *g*) for 1 min. The lower layer of liquid was poured into a 2 mL collection tube and then centrifuged (4 °C, 10,000× *g*) for 3 min. The RB column was placed into a new microcentrifuge tube, and 50 µL of RNase-free water was added to absorb RNA. The mixture was allowed to stand for 5 min and then centrifuged (4 °C, 10,000× *g*) for 2 min to obtain purified RNA. Subsequently, a NanoDrop (MaestroGen, Hsinchu City, Taiwan) was used for analysis. The ratio of OD_260_ to OD_280_ was measured using a spectrophotometer to assess the concentration of purified RNA. The ratio of 1.9–2.0 indicates high purity RNA. The RNA extracts were stored at −80 °C for later use.

### 2.4. Transcriptome High-Throughput Next-Generation Sequencing

A 10 µg total RNA sample was sent to a sequencing service company for RNA sample quality testing to confirm the integrity of the RNA sample. The use of contaminated or degraded samples was avoided. Only high-quality total RNA was selected, and these RNA samples were divided into two groups that include CT and CS. Tissues (brain, gill, liver, and muscle) were each added to one specific tube, and total RNA extracts from CT (*n* = 6) and CS (*n* = 6) brain, gill, liver, and muscle tissue samples were used for transcriptome gene library construction according to high-throughput next generation sequencing (NGS) platform. The transcriptome assembly of eight libraries was submitted to the NCBI short read archive database (accession number: SUB 9446960). High-quality read sequence data for unigene assembly were used, and they were screened for differentially expressed genes and utilized for Gene Ontology (GO) annotation and pathway enrichment analysis. Additionally, Kyoto Encyclopedia of Genes and Genomes (KEGG), Clusters of Orthologous Groups (COG) annotation and prediction analyses of encoded proteins were performed using gene function annotation. MicroSAtellite (MISA) was used to search for simple sequence repeat (SSR) or microsatellite DNA markers on the reference sequence of the transcript. Primer3 (v2.3.7) [27] was used to analyze the genes for the detected SSR markers and for the design of the primer pair for the locus. HISAT (v0.1.6) software [28] was used to compare clean reads for genes, and GATK (v3.4-0) [29] was used to detect single nucleotide polymorphisms (SNPs) and to filter low-quality SNPs.

### 2.5. Transcript Database Gene Differential Expression

From the Nile tilapia transcript gene library, the maximum and minimum log2 ratio (CT/CS) 2 of the expression of the cold-tolerance group (CT) and the cold-sensitivity group (CS) was selected. Digitized gene expression (DGE) analysis and selection of genes exhibiting differential expression were conducted. Gene differential performance analysis was used to identify genes possessing different expression levels in different samples, and their expression levels according to FPKM were expressed.

Fragments per kilobase of exon per million fragments mapped (FPKM) was calculated according to the following formula:$$\mathrm{FPKM}=\frac{total\;exon\;reads}{mapped\;reads\left(millions\right)\times exon\;length\left(KB\right)}$$

Total fragments represent the number of fragments that are uniquely aligned to the gene, mapped reads represent the total number of fragments that are uniquely aligned to all genes, and exon length is the number of bases within the gene [30].

### 2.6. Reverse Transcription Polymerase Chain Reaction

A high-capacity cDNA reverse transcription kit (Applied Biosystems—Life Technologies, Carlsbad, CA, USA) was used for reverse transcription. The total reaction volume was 20 µL, and the components were 1 µg of RNA, 2 µL of 10× RT buffer, 0.8 µL of 25× dNTP mix (100 mM), 2 µL of 10× RT Random primer, 1 µL of MultScribe^TM^ reverse transcriptase (50 U/µL), and 4.2 µL of Nuclease-free water. Reaction was performed using Veriti^®^ thermal cycler (Applied Biosystem—Life Technologies), and the reaction conditions were 25 °C for 10 min, 37 °C for 120 min, and 85 °C for 5 min. After the reaction, the sample was diluted 100-fold and stored at −20 °C for subsequent use.

### 2.7. Real-Time Quantitative Polymerase Chain Reaction

To verify the credibility of the RNA*seq* results, six functional genes (CL10781_10, CL1487_25, CL5212_1, CL5902_1, Unigene196, and Unigene7071) were used for qPCR expression analysis and screened based on: (1) expression ability in four tissues, including brain, gill, liver, and muscle; (2) differences in expression between the cold-tolerance group (CT) and the sensitive group (CS); (3) the presence of SSR or SNP molecular markers. The tested samples of cold-tolerance (CT, *n* = 6) and cold-sensitivity (CS, *n* = 6) previously used for RNA*seq* were used for qPCR analysis. The total reaction volume was 20 µL, and the components were 10 µL of 2× Power SYBR^®^ Green PCR Master Mix, 5 µL of cDNA (diluted 100-fold), 1.2 µL of forward primer (0.5 µM), 1.2 µL of reverse primer (0.5 µM) and 2.6 µL of RNase-free water. A Roche LightCycler^®^ 480 Real-Time PCR System (Roche Applied Science, Mannheim, Germany) was used for the reaction. The reaction conditions were 50 °C for 2 min in the first stage, 95 °C for 10 min in the second stage, and 40 cycles at 95 °C for 15 s and 60 °C for 30 s in the third stage. The obtained Ct value was used as the mean and standard deviation (SD), and the data from the control group (18S rRNA and β-actin) were deducted as a relative quantification expression graph. The primer pairs of candidate target and internal reference genes employed are listed and shown in Appendix A.

### 2.8. Genomic DNA Extraction

Approximately 100 mg of the tail fin of the test fish was cut and placed into a 1.5 mL microcentrifuge tube containing 800 µL of 70% ethanol solution. Information regarding the source of the sample that included the sample number, sample name, and sampling date was attached to each centrifuge tube. After computer labeling, the samples were stored at −20 °C and frozen for genomic DNA extraction. The MasterPure^TM^ DNA Purification Kit (Epicenter, Madison, WI, USA) was used to extract genomic DNA from Taiwan tilapia. The fin sample was placed into a new 1.5 mL microcentrifuge tube, and 800 µL of tissue and cell lysis solution (Epicenter) and 2 µL of proteinase K (Epicenter) were both added. The samples were mixed well and placed into a 55 °C oven for 6–12 h. After the tissue was completely dissolved, 400 µL was pipetted into a new centrifuge tube, and 250 µL of MPC protein precipitation reagent (Epicenter) was added and mixed thoroughly. The mixture was centrifuged at 10,000× *g* and 4 °C for 10 min. Thereafter, 500 µL of supernatant was pipetted into a new 1.5 mL microcentrifuge tube, and 500 µL of isopropanol (Sigma-Aldrich, Saint Louis, MO, USA) was added and mixed several times to obtain dehydrated DNA pellets. The pellets were centrifuged to the bottom of the tube using a microcentrifuge (Spin mini), and the supernatant was poured out. Next, 100 µL of 70% alcohol was added to the 1.5 mL microcentrifuge tube for two washes. After centrifugation to remove residual alcohol, the tube was placed in an oven at 55 °C for 10 min to completely dry the alcohol. Subsequently, 200 µL of ddH_2_O was added, and the tube was placed into a dry bath at 37 °C for 20 min to dissolve the DNA.

A Maestro Nano Spectrophotometer (Maestrogen, Las Vegas, NV, USA) was used to measure the OD_260_ and OD_280_ absorbance values and to calculate the DNA concentration. Then, 0.8% agarose gel electrophoresis was used for electrophoresis separation. GelRed Nucleic Acid Gel Stain (Biotium, Inc., Fremont, CA, USA) was used after staining for 30 min, and the Slite140 Compact Gel Documentation System (Avegene Life Science, New Taipei City, Taiwan) colloid camera system was used to assess the quality of genomic DNA. After measuring the computer label of the sample source information that includes tissue number, sample name, extraction date, and other information, the samples were stored in a frozen state at −20 °C for subsequent testing.

### 2.9. Microsatellite Marker DNA Genotyping

The microsatellite markers that were developed using two low-temperature tolerance-related microsatellite markers and 13 transcript functional databases targeted 269 low-temperature-tested Taiwan tilapia (RF, *n* = 120; YH, *n* = 102; NT, *n* = 47) for genotyping analysis. Based on the multiple fluorescent labeling method, the first PCR amplification used the forward primer containing the adaptor to bind gDNA fragments. The total volume of the reaction solution was 10 µL, and the composition was 5 µL of 2× Taq DNA polymerase Mastermix (Bioman, Taipei, Taiwan), 0.3 µL of forward primer, 0.3 µL of reverse primer, 2 µL of template DNA, and 2.4 µL of ddH_2_O. PCR was performed using a 96-well polymerase chain reactor (Veriti^®^ thermal cycler; Applied Biosystems—Life Technologies). The reaction conditions were 94 °C for 3 min, 94 °C for 45 s, annealing temperature (*T**_a_*) °C for 30 s, 72 °C for 30 s, and 72 °C for 7 min. The four steps were cycled 30 times. When the second PCR amplification was performed, the forward primer was changed to fluorescent primers that included JOE (green), FAM (blue), ROX (red), NED (yellow), and four different fluorescent primers. After labeling, the total volume of the reaction solution was 10 µL, and the components were 5 µL of 2× Taq DNA polymerase Mastermix (Bioman), 0.3 µL of fluorescent primer, 0.3 µL of reverse primer, 2 µL of the first PCR product diluted 10-fold as template DNA and 2.4 µL of ddH_2_O. A 96-well polymerase chain reactor (Veriti^®^ thermal cycler; Applied Biosystems—Life Technologies) was used for PCR, and the reaction conditions were the same as those used for the first PCR amplification.

After the PCR products were separated using electrophoresis with a 2% agarose colloid, they were stained with GelRed Nucleic Acid Gel Stain (Bioman) for 30 min, photographed, and assessed using the Slite 140 Compact Gel Documentation System (Average Life Science, New Taipei City, Taiwan) colloid camera system. The probability of allele identification errors was reduced, and each sample was subjected to a second PCR analysis. If the same result was obtained twice, the microsatellite genotype of this sample was determined. After mixing four different fluorescently labeled PCR products, 5 µL of these products were further mixed with 0.2 µL of GeneScan^TM^-500 LIZ^TM^ size standard (Applied Biosystems, Foster City, CA, USA) and 10.8 µL HiDi^TM^ Deionized formamide (Applied Biosystems, Foster City, CA, USA) and centrifuged. After heating at 94 °C for 3 min, the samples were quickly placed on ice for 5 min and then placed on an ABI PRISM^®^ 3730xl DNA Analyzer (Applied Biosystems, Foster City, CA, USA) automatic DNA analyzer for short tandem repeat (STR) capillary electrophoresis separation. The obtained microsatellite marker data were represented by the analogy of A, B, C, and so forth according to the size of the allele fragments, using Geneious software for the interpretation and analysis of multiple fluorescent PCR polymorphic marker genotypes.

### 2.10. Single Nucleotide Polymorphism Markers Genotyping

The principle of typing is to perform a single-base extension reaction at the SNP site and then use mass spectrometry to analyze the molecular weight of the product. As a single base extends four bases (A, T, C, and G), the molecular weights are all different (ddATP: 475.18 g/mol, ddCTP: 451.16 g/mol, ddGTP: 491.18 g/mol, ddTTP: 466.17 g/mol), and SNP genotypes can be analyzed based on this feature. SNP stereotype ratio, HM call, and LM call respectively represent the ratio of homozygous high-molecular-weight base type to low molecular weight base type. For example, when assessing at SNP of the mutual conversion of cytosine (C) and thymine (T), the molecular weight of the T base is greater than is the molecular weight of the C base. Based on this, the T base will be set as high mass (HM), and the C base will be set as low mass (LM).

Thirty-seven single-nucleotide polymorphism gene-based markers were selected from the transcript database for use in the targeting of 190 Taiwan tilapia populations (RF = 72, YH = 96, NT = 22) that were tested under low temperature using the Sequenom MassARRAY platform and iPLEX gold chemistry (Sequenom, San Diego, CA, USA) for genotyping analysis. Design-specific PCR primers and extension primers created using the Assay Designer software package (v.4.0) (Premier Biosoft International, Palo Alto, CA, USA) were used to configure a total volume of 5 µL of the reaction solution for multiplex PCR reactions. The composition was 1 µL of genomic DNA (10 ng/μL), 0.2 units of *Taq* polymerase, 2.5 pmol of PCR primer, and 25 mM of dNTP (Sequenom, San Diego, CA, USA). PCR reaction conditions were 94 °C for 4 min, 94 °C for 20 s, 56 °C for 30 s, 72 °C for 1 min, and 72 °C for 3 min. Steps 2–4 were cycled 45 times, and 0.3 U of shrimp alkaline phosphatase (SAP) was added to inactivate the excess dNTPs. The iPLEX enzyme, terminator mix, and extension primer (Sequenom, San Diego, CA, USA) were used for the single-base extension reaction. The PCR reaction conditions were 94 °C for 30 s, 94 °C for 5 s, 56 °C for 5 s, 80 °C for 5 s, and 72 °C for 3 min. Steps 3–4 were cycled five times, and then the program transitioned back to Step 2 for 40 cycles. A cation exchange resin was added to remove the residual salts. Then, 7 nL of the product was placed on the 384-well SpectroCHIP (Sequenom, San Diego, CA, USA), and a MassARRAY Analyzer 4 was used for analysis. Then, TYPER 4.0 analysis software (http://sequenom.com/Genetic-Analysis/Applications/iPLEX-Genotyping/iPLEX-Literature, accessed on 15 October 2021) was used to read the mass spectrometry results for data analysis and SNP genotype result interpretation. The genotype call score parameter analysis result determines if it can be interpreted as a reliable result. The genotype call score calculation method was as follows:Genotype call score = P_MA_ × P_YLD_ × P_SKW_

The genotype call score was calculated using three parameters that include the signal intensity parameter of the PYLD-peak, the molecular weight accuracy and signal sharpness of PMA-pea, and the signal intensity ratio of the two genotypes of PSKW-SNP. The software determines if each peak in the mass spectrum is a credible result and performs SNP interpretation based on these results. When the genotype call score is lower than the cut-off value that the software can trust, no genotype interpretation will be provided. If the genotype call score is above the cut-off value, the software can perform genotype interpretation. According to the genotype call score from low to high, the software will be aggressive, moderate, or conservative in the final analysis result as a reference.

### 2.11. Statistical Analysis

The obtained genotype data were imported into PopGene32 software [31] to statistically analyze the number of alleles (*Na*) and allele frequency (*Ne*) of each microsatellite locus, and the various evaluating coefficients of genetic diversity and miscellaneous data were observed. Observed heterozygosity (*H_o_*), expected heterozygosity (*H_e_*), polymorphism information content (*PIC*), and fixation index (*F_IS_* and *F_ST_*) were calculated as follows.

Observed heterozygosity (*H_o_*):*H_o_* = *N_het_* ÷ (*N_hom_* + *N_het_*)
where *N_het_* is the number of heterozygous individuals, and *N_hom_* is the number of homozygous individuals.

Expected heterozygosity (*H_e_*):$$H_{e}=1-\sum_{j}^{n}Pi^{2}$$
where *n* is the number of alleles at each locus, and *Pi* is the frequency of the *i*th allele [32].

Polymorphism information content (*PIC*):$$PIC=1-\sum_{i=1}^{k}\;pi^{2}-\sum_{i=1}^{k-1}\sum_{j=i+1}^{k}2pj^{2}pj^{2}$$
where *k* is the number of alleles, and *pi* and *pj* are the gene frequencies of the *i*th and *j*th alleles [33].

Fixed index (*F_IS_*):*F_IS_* = 1 − *H_o_* ÷ *H_e_*
where *H_o_* is the observed heterozygosity, and *H_e_* is the expected heterozygosity [34].

The phenotypic allele effect of SSR/SNP which was associated with cold-tolerance trait was estimated through comparison between the average phenotypic values over accessions with the specific allele.

The IBM SPSS Statistics version 22.0 program (SPSS Inc., Chicago, IL, USA) was used for statistical analysis of the association between the phenotypes and the markers. According to ANOVA and Scheffe’s 95% confidence level (post-hoc test), genotype and body phenotype differences were significant.

## 3. Results

### 3.1. Phenotypic Differences in Cold-Tolerance as Assessed by the Loss of Balance Behavior in Response to Cooling Stress

The results from the reduction test indicated that RF fish only reduced their food intake when the water temperature reached 18.3 °C, and only a small number of these fish would eat food at a water temperature of 12.3 °C. The fish began to roll over at 10.5 °C. At 8.6 °C, the rollover rate reached 55.5%, and at 7.7 °C, the rollover rate became 100%. YH fish only consumed less food when the water temperature reached 19.0 °C, and only a small number would eat when the temperature reached 13.3 °C. At 11.2 °C, the fish began to roll over. At 9.3 °C, the rollover rate reached 51.9 %, and at 8.0 °C, the rollover rate became 100%. The NT fish only consumed less food when the water temperature reached 18.7 °C, and only a small number of fish would eat at 12.2 °C. The fish began to roll at 11.5 °C. At 10.3 °C, the rollover rate reached 50.1%, and at 9.3 °C, the rollover rate became 100%.

Among the three low-temperature test groups, the RF Taiwan tilapia group exhibited the best cold-tolerance with an average temperature tolerance of 8.74 ± 0.55 °C, and this was followed by the YH Taiwan tilapia group with an average temperature tolerance of 9.36 ± 0.72 °C. The NT Taiwan tilapia population possessed the worst low-temperature tolerance with an average tolerance temperature of 10.16 ± 0.45 °C (Figure 1).

### 3.2. Transcriptome Sequencing Analysis Overview

#### 3.2.1. RNAseq Retrieval, Pre-Processing, Assembly, and Annotation of the Unigenes

In this study, the cold-tolerant fish (those that can tolerate 8.74 ± 0.55 °C) exhibit lower temperature tolerance than do the cold-sensitive group of fish (that can tolerate 10.16 ± 0.45 °C). The collection of tissue samples used for transcriptome sequencing was based on the lowest tolerable temperature (10 °C) of the cold-sensitive group. Concurrently, cold-tolerant group fish samples were collected. The differences in measured gene expression were indeed caused by the differences in tolerance.

Using the Illumina Hiseq 2000 next-generation platform, the transcriptome gene library was constructed for the total RNA of the brain, gill, liver, and muscle tissues of the two groups of cold-tolerance (CT) and cold-sensitivity (CS) fish. 

After quality trimming and filtering of low-quality reads, 234,765,554 high-quality paired-end (PE) reads were generated (each with a length of 150 bp), and a total of 35,214,833,100 nucleotides of data (Table 1) were generated for assembly from scratch.

The length distribution statistics of the unigene transcripts after assembly are presented in Appendix A. A total of 128,147 unigenes were obtained, and these possessed a total length of 185,382,926 nt, an average length of 1446 nt, an N50 of 3157 nt, and a GC content of 47.46% (Appendix A).

Unigene was used to compare the distribution of genes based on blastx NR annotation, and the species comparisons utilized *Oreochromis niloticus*, *Haplochromis burtoni*, *Neolamprologus brichardi*, *Maylandia zebra*, and other species (Figure 2). The analysis results revealed that 48,521 single genes exhibited higher homology and the highest similarity to *Oreochromis niloticus*, where they accounted for 64.99% of the total. By comparing unigenes to the nucleic acid database Nt (*p*-value < 0.00001) through the use of blastn, the protein possessing the highest sequence similarity to that of the unigenes can be determined, and information regarding the protein function of the unigene can be obtained. A total of 71,890 predicted CDS were compared to listings in the protein database, and 1389 predicted CDS were identified for a total of 73,279.

In terms of function annotation, unigenes included protein function annotation and COG function annotation. Unigenes were annotated to the NR, NT, Swiss-Prot, COG, GO, and KEGG databases (Appendix A), and we counted the number of unigenes annotated to each database and also the total number of annotations (Table 2). The cross-comparison analysis of the Venn diagrams of the four major protein databases yielded 25,844 co-annotated unigenes (Figure 3).

Based on the COG database, a large proportion of sequences within the transcriptome of Taiwan tilapia fish participate in the functional classification of “general function prediction only”. Additionally, “replication, recombination, repair”, “transcription”, “signal transduction mechanisms”, “cell cycle control, cell division, chromosome partitioning” and “translation, ribosomal structure and biogenesis” pathways were identified (Appendix A).

The functional classification of these transcripts according to the GO database indicates that “cellular process”, “single-organism process” and “metabolic process” are the primary involved functions. Biological processes such as “cell”, “cell part”, “membrane” and “membrane part” are the primary cellular components involved, and “binding” and “catalytic activity” are the primary molecular functions involved (Appendix A).

#### 3.2.2. Detection of Microsatellites and Single Nucleotide Polymorphism Markers

Using the assembled unigene as the reference sequence, the MicroSAtellite (MISA) software was used to search for a total of 38,377 microsatellite markers containing one (single), two (double), three, four, five, and six base repeats. The analysis identified 8745 single-base repeats, 17,610 double-base repeats, 10,045 three-base repeats, 1182 four-base repeats, 650 five-base repeats, and 145 six-base repeats (Figure 4A).

Additionally, after comparing clean reads to genes using the HISAT (v0.1.6) software, GATK (v3.4-0) software was used to search for single nucleotide polymorphisms (SNPs) and to filter low-quality SNPs. There were 28,093 and 27,746 identified transitions for adenine (A)/guanine (G) and cytosine (C)/thymine (T), respectively. There were also 5783, 5437, 6444, and 5822 identified transversions of adenine (A)/cytosine (C), adenine (A)/thymine (T), cytosine (C)/guanine (G), and guanine (G)/thymine (T), respectively (Figure 4B).

### 3.3. Transcriptome Responses to Temperature Decreases

#### 3.3.1. Differential Gene Expression between Cold-Tolerant and Sensitive Fish

To compare the cold-tolerance (CT) and cold-sensitivity (CS) groups of Taiwan tilapia in the context of the brain, gill, liver, and muscle (CT-B vs. CS-B, CT-G vs. CS-G, CT-L vs. CS-L, and CT-M vs. CS-M), differential expression of transcripts was assessed. Parameters that included a false discovery rate (FDR) < 0.05, log_2_ fold change > 1, or log_2_ fold change < − 1 were used to screen for differentially expressed genes (DEGs) from the DE-*seq* analysis. All unigenes represent the results for up-regulation and down-regulation in regard to differential expression distribution based on quantitative analysis (Figure 5). Totals of 4191, 4235, 3164, and 2439 differentially expressed genes were detected in the brain, gill, liver, and muscle tissues, respectively. The numbers of regulatory genes within the four tissues of the cold-tolerance (CT) group were 2081, 2261, 1824, and 1200, respectively. There were 2110, 1974, 1340, and 1239 downregulated genes, respectively (Figure 5A,B). The results revealed that the low-temperature treatment process resulted in significant differences in gene expression between cold-tolerance (CT) and cold-sensitivity (CS) fish.

From a total of 10,380 differentially expressed genes, the Venny online software (2.1) (https://bioinfogp.cnb.csic.es/tools/venny/, accessed on 15 October 2021) was used to cross-check CT-B vs. CS-B, CT-G vs. CS-G, CT-L vs. CS-L, CT-M vs. CS-M, and other. Venn diagram analysis of upregulated and downregulated genes in the four tissues revealed that 156 (1.5%) genes could overlap among the four tissues (Figure 5C).

#### 3.3.2. Differential Expression of Functional Genes Containing SSRs and SNPs

Among the 156 genes exhibiting differential performance in the four tissues described above and combined with the microsatellite and single nucleotide polymorphic marker database previously explored from transcripts, from the 38-cold tolerance-related functional genes that were identified using differential gene expression analysis in the four tissues, 13 genes with 13 microsatellites (Table 3) and 25 genes with 37 single nucleotide polymorphism markers (Table 4) were identified. Of these, six unigenes contained two to five SNP markers. Each unigene containing SSR and SNP markers corresponded to the observed expression level (Figure 6).

MicroSAtellite (MISA) and GATK (v3.4-0) software programs were used to label the microsatellite (SSR) and single nucleotide polymorphism (SNP) variant sequences and to determine other biological information regarding the 156 differentially expressed genes selected above. The analysis revealed that there were 13 and 25 genes possessing SSR and SNP sequences, respectively. The online software ClustVis was used to perform pattern clustering analysis of the FPKM value of the marker gene and to create a heat map to indicate the gene difference performance (Figure 6A,B).

Additionally, based on the results of the gene annotation database and the NCBI BLAST tool used to compare DNA sequence data, Sequence Viewer was used to annotate genes to the RefSeq database, where 13 candidate SSRs (Figure 6A) and 37 candidate SNPs were marked (Figure 6B). Gene function annotations located in gene exons, introns, 3′- and 5′-untranslated regions (UTR), and other positions are listed in Table 3 and Table 4, respectively.

#### 3.3.3. Validation of the Transcriptome Sequencing Results Using Real-Time qPCR

To verify the results obtained by RNA*seq*, six genes were randomly selected for qPCR analysis. The results revealed that the expression and transcription levels of RNA*seq* and qPCR genes in the brain, gill, liver, and muscle tissues of the cold-tolerance (CT) and cold-sensitivity (CS) strains were similar for each gene, and the two data sets were highly correlated (R^2^ = 0.9794, *p* < 0.001, Figure 7). This result indicates that the RNA*seq* results were reliable.

In this experiment, the results of target gene sequence amplification revealed that the number of molecules can be doubled in each replication cycle and that this process exhibits excellent amplification efficiency, thus indicating that the primer design of this experiment is appropriate and of high quality and that there are no problems with secondary structures. The optimal qPCR reagent concentration and reaction conditions yielded ideal results.

### 3.4. Correlation between the Genotypes of the Polymorphic RNAseq Markers and Cold-Tolerance with Significant Genetic Variation in Taiwan Tilapia

#### 3.4.1. Identification of Candidate SSR Markers Involved in Cold-Tolerance

Using 13 microsatellite markers to analyze the diversity of the three Taiwan tilapia populations (RF, YH, and NT), it was determined that the Unigene196 marker can produce the largest number of alleles (16) and genotype combinations (44), while CL1876_16 and CL279_7 yielded only two allele numbers and two genotype combinations (Table 5). Previous studies have developed two cold-tolerance markers that are identified as UNH916 and UNH999 [19]. According to the test results of this experiment, it was determined that UNH916 and UNH999 can produce 12 and 14 alleles in the Taiwan tilapia test population, respectively. Among them, the RF population possessed the two markers with the most alleles (9 and 12, respectively), and the NT population possessed the least alleles (2 and 3). The E, H, J, and L alleles marked by UNH916 were only observed in the RF population, and allele I was only observed in the YH population. The I and N alleles marked by UNH999 were only observed in the RF and YH populations, respectively (Appendix A).

The number of alleles (*N_a_*) was 2–13, 1–12, and 1–4, and the number of genotypes (*N_g_*) was 2–24, 1–20, and 1–5. The average observed heterozygosity (*H_o_*) values were 0.535 ± 0.27, 0.448 ± 0.26, and 0.347 ± 0.31, and the average expected heterozygosity (*H_e_*) values were 0.595 ± 0.23, 0.449 ± 0.26, and 0.293 ± 0.26, respectively. The average polymorphism average information contents (*PIC*) were 0.591 ± 0.22, 0.447 ± 0.26, and 0.290 ± 0.25, respectively. The average fixed index (*F_IS_*) values were 0.178 ± 0.28, −0.008 ± 0.14, and −0.210 ± 0.17 (Table 5).

IBM SPSS Statistics v22.0.0 and the Scheffe method (SPSS Inc., Chicago, IL, USA) both were used to analyze the genotypes and to verify the correlation with the minimum temperature tolerance of 13 microsatellite markers in the three Taiwan tilapia populations (RF, YH, and NT). The results revealed that myosin-10 isoform X3 (CL1876_16), R3H domain-containing protein 1 (CL279_7), protein IWS1 homolog isoform X3 (Unigene7071), bifunctional methylenetetrahydrofolate dehydrogenase/cyclohydrolase, mitochondrial-like (CL9318_1), and other functional gene markers were significantly associated with the cold-tolerance traits of Taiwan tilapia (*p* < 0.05) (Table 5).

#### 3.4.2. Genotype of the Gene-Based SNP Marker That Was Significantly Correlated with Cold-Tolerance

The results of the SNP assay analysis of 37 candidate single nucleotide polymorphism markers related to low temperature tolerance traits revealed that the success rate of SNP marker gene typing was as high as 99.30%, and the reliability (conservative calls) averaged 87.98% (Table 6). The average proportions of homozygous and heterozygous genotypes were 83.8% and 13.5%, respectively. The AutoCluster model was used to classify homozygote and heterozygote according to the population characteristics of the samples acquired from different Taiwan tilapia populations. The peak signal was used as the coordinate axis to draw a two-dimensional graph, and the genotype was then judged based on the clustering results (Figure 8). Among the 37 single nucleotide polymorphism markers, 26, 20, and four markers in the RF, YH, and NT Taiwan tilapia populations, respectively, were polymorphic, and a total of 27 markers were polymorphic among the three test populations (Table 6).

IBM SPSS Statistics v22.0.0 and the Scheffe test method (SPSS Inc., Chicago, IL, USA) were both used to analyze the correlation between the genotypes of 37 single nucleotide markers and the minimum tolerable temperature of the three Taiwan tilapia populations (RF, YH, and NT). The results revealed the SNPs in the transcript functional database. The gene marker CL5212_1 was significantly correlated with the low-temperature tolerance of the YH Taiwan tilapia population (*p* < 0.01) (Table 6).

### 3.5. Verification of the SNP Marker Assisted Selection for Cold-Tolerant Strains in Taiwan Tilapia

CL5212_1 marker from CT and CS species were used to select parental individuals possessing AA and dd genotypes, respectively, and to perform CS × CS, CS × CT, CT × CS, CT × CT positive and negative hybridization pairings to produce SC1 (*n* = 87), SC2 (*n* = 148), SC3 (*n* = 172), and SC4 (*n* = 110) (Figure 9). The progeny fish of the four families were subjected to low-temperature tolerance test, and the minimum temperature tolerance of each fish was recorded. Genomic DNA was extracted from the collected fin samples to confirm the genotype. Significant differences in cold-tolerance were observed in the four tested hybrid families (*p* < 0.01) that originated from the SC1 family with homozygous A/A cold-tolerant genotypes, and the lowest temperature tolerance was 7.3 ± 0.23 °C. For SC2 and SC3 family fish possessing the heterozygous A/d genotype, the minimum tolerable temperatures were 8.35 ± 0.49 °C and 8.47 ± 0.31 °C, respectively. For the SC4 family fish possessing the homozygous d/d cold-sensitive genotype, the minimum tolerable temperature was 10.47 ± 0.70 °C. This study demonstrates that the CL5212_1 marker is a selection marker that can be used for a marker-assisted breeding platform.

## 4. Discussion

Tilapia is an important farmed fish species in tropical and subtropical regions. It is often exposed to prolonged or repeated cold stress at low temperatures during winter, and this stress can result in a reduction or a complete halt of food intake. Physiological responses such as stunting and death can seriously affect yield and quality [7,18,35,36]. Sensitivity to low temperatures is the primary limiting factor that hinders the introduction and distribution of tilapia in temperate and high-elevation areas.

Since the late 1990s, several Nile tilapia strains exhibiting low temperature adaptability have been domesticated and cultivated [37,38]. Understanding the genetic basis for intra-species phenotypic variation is a long-standing goal in biological breeding [39,40]. In addition to providing abundant genetic resources, body phenotypic variation that has undergone long-term evolution or multi-generational improvement also plays a key role in the contribution of gene expression pattern variation [41]. Research comparing the cold-tolerance characteristics among the different Nile tilapia species located in different geographic locations such as Egypt, Ghana, and the Ivory Coast revealed that Nile tilapia have survived through natural selection for many years, and the physiologies and lethal temperature ranges reflect different cold-tolerant phenotypic characteristics acquired from their ancestral populations [42,43]. Although it is well known that there are intraspecies phenotypic differences in temperature tolerance [44,45,46,47], there is still a lack of information regarding this genetic variation. The physiological basis of this study was the long-term collection of populations possessing different cold-tolerant phenotypes. After multiple generations of low-temperature pressing and parental family selection, cold-tolerant and cold-sensitive strains were established.

Since 2007, the Taiwan tilapia research team has been cooperating with the Fisheries Research Institute of the Executive Yuan Agriculture Committee to perform genetic improvement studies for the selection of growth traits for Nile tilapia broodstock [25,26]. In addition to the pure Nile strain (code name NT), samples of a variety of Taiwan tilapia species were continuously collected from northern Taiwan (Rui-Fang wild field, code RF) and southern Taiwan (private commercial seed breeding farm, code YH). These fish are maintained in the core breeding greenhouse of the Department of Aquaculture at National Taiwan Ocean University and the Aquatic Organism Research and Conservation Center in Gongliao. In addition to establishing the family code of each full-sib family through the use of the Avid Identification System, each strain also tracks the performance of each family in F1 and F2 for different generations of cold tolerance trait selection. These characteristics were measured from individual fish that survived in the F1 and F2 generations. According to the previous results involving cold tolerance data, the cold tolerance of the two generations from the same sibling family exhibits hereditary performance.

A number of studies have used high-throughput next-generation sequencing technology to examine Atlantic salmon [14], channel catfish (*Ictalurus punctatus*) [20], zebrafish (*Danio rerio*) [21,23,48], Nile tilapia (*Oreochromis niloticus*) [24,49,50,51], Blue Tilapia (*Oreochromis aureus*) [9], Japanese flounder (*Paralichthys olivaceus*) [52], carp (*Cyprinus carpio haematopterus*) [22], milkfish (*Chanos chanos*) [53], rainbow trout [54], and other tropical and temperate aquaculture models of fish for low-temperature domestication-related regulatory mechanism research [48,55]. However, apart from systematically comparing gene expression patterns of specific tissue transcripts of the same fish species possessing different low-temperature tolerance genetic mutations, there is limited information regarding functional genes that can allow for molecular marker-assisted breeding.

In the present study, transcriptomic gene libraries of the cold-tolerant (CT) and cold-sensitive (CS) groups were constructed using high-throughput next generation sequencing (NGS). A total of 48 total RNA samples (2 groups × 6 fish/group × 4 tissues/fish) were extracted from four tissues that included the brain, gill, liver and muscle. Biological analysis met the requirements of the test sample repetition. Additionally, research reports by Assefa et al. [56] and Zhao et al. [57] noted that pooling of RNA samples can not only reduce the cost of sequencing (optimize the cost) but also maintain high-throughput sequencing and high quality (maintaining the power). In this study, based on the consideration of sequencing cost, we mixed high-quality total RNA samples for the same tissue in the same group of six samples and then sequenced them, and this produced high-quality sequencing results.

This study proposes for the first time that the Illumina HiSeq 2000 platform can be used to observe the transcriptome of the brain, gill, liver, and muscle tissues of the two strains of Taiwan tilapia under low temperature stimulation in the context of genetic research. Among the identified unigenes, 100,108 (78.12%) were successfully annotated in the public NR, GO, COG, KOG, and KEGG databases through the use of a BLAST search. GO and COG analyses determined the distribution of functional genes in tilapia from Taiwan. The KEGG database search successfully revealed the functions of cellular process genes and the gene products of metabolic processes.

Low temperature and cold adaptation culture model fish species such as Japanese flounder (*Paralichthys olivaceus*) [52] and common carp (*Cyprinus carpio haematopterus*) [22] have been previously studied, and the transcriptome KEGG analyses revealed that the differentially expressed genes primarily participate in cell pathways (intercellular communication, cell movement, membrane transport, and catabolism), signal transmission (signal molecular interaction, genetic information processing, folding, classification, degradation, transcription and translation), metabolism (amino acid metabolism, lipid metabolism, energy metabolism), and the biological system (endocrine, circulation, digestion, nerve, sensory, excretion and immune system). Additionally, transcript sequencing analysis examining milkfish (*Chanos chanos*) used transcript sequencing and determined that this low-temperature sensitive fish species can produce enough energy to resist cold under low temperature stress, and the glycogen and fatty acid degradation- and synthesis-related functional genes appear to be up- and down-regulated, respectively [53].

The KEGG analysis in this study revealed that the metabolic pathways exhibit common enrichment in four tissues, including the brain, gill, liver, and muscle. Among these, tight junctions are the most enriched pathways in the brain and gills, and these structures support the continuous intercellular barrier between epithelial cells that separates interstitial spaces and regulates the selective passage of solutes through epithelial cells [58]. In liver tissues, protein degradation (ubiquitin-mediated proteolysis) that is primarily guided by ubiquitin, is the most enriched pathway. It is extensively involved in the regulation of the cell cycle, immune and inflammatory responses, signal control transmission pathways, and development and differentiation [59]. The most enriched pathway in muscle tissue is the MAPK signaling pathway that uses protein networks to transmit extracellular signals to the cell to regulate cellular processes. It involves the activation of cell membrane signaling molecules and protein kinases [60]. It is noteworthy that this reflects the interactions between the cell and the surrounding environment, adjacent cells, and ECM, and this pathway participates in intracellular processes, such as protein processing and RNA transport and degradation. These findings are consistent with the results of previous transcriptomic analyses [9,22,48,52].

The current study used the qPCR test and the 18S rRNA and β-actin as the internal references genes, which were quite stable. Based on repeated test results, the average Ct values for the 18S rRNA and β-actin mRNAs were determined to be 16.96 ± 0.24 and 15.06 ± 0.47, respectively. In several previous research studies, with *Oreochromis niloticus* [26,48,51], *Oreochromis aureus* [9] and *Paralichthys olivaceus* [52], the 18S rRNA and β-actin genes were also used and proved as good internal reference expression genes.

Molecular markers can be used to evaluate the genetic diversity and genetic purity of a population. A significant reduction in genetic diversity causes the population to decline in regard to close relatives and affects the performance of phenotypic traits [61]. Regarding the genetic diversity of tilapia populations and the development of microsatellite markers, Zhu et al. [19] screened UNH916 from the genomic DNA of cold-tolerant and cold-sensitive fish. With the UNH999 microsatellite marker locus, the allelic fragment amplified through the use of primer pairs can be used for family codominance analysis and identification and also to assess polymorphisms. Similar results were obtained for the two sets of SSR markers in the three test populations used in this study. The total number of genotype combinations (*n* = 52) generated was more than 2-fold the total number of alleles (*n* = 25). However, it is speculated that this study may be affected by the experimental species and the number of analyses, and these factors may result in an inability to identify the same cold-tolerant allele from the two groups of microsatellites. Additionally, eleven sets of novel SSR markers that were developed from transcripts were added to analyze the correlation between the diversity index and cold tolerance traits. By analyzing the diversity of multiple sets of alleles in addition to using genetic parameters such as *H_o_* (0.347–0.535), *H_e_* (0.293–0.595), and *PIC* (0.920–0.591) to demonstrate that Taiwan tilapia breeding populations still retain a high genetic polymorphism rate within the genomic DNA, researchers can also develop functional SSR markers for cold-tolerant fish strains. This can assist the seedling industry in the future to achieve scientifically improved germplasm for cold-tolerance molecular breeding purposes.

Due to the large number of SNPs and their widespread distribution within the genome, they provide an advantage in terms of quantitative trait locus (QTL) mapping and are crucial molecular markers for genetic research [62,63]. In the development of fish SNP markers, many research teams have used simplified genome sequencing methods to develop markers that can be used for genotyping and selection of different geographic locations and strains [64]. The SNP markers developed through the use of RNA*seq* and RAD-*seq* [65,66] are not only used for genetic diversity analysis in GIFT, *O. niloticus*, *O. mossambicus*, red tilapia and other ethnic groups but have also been used to develop and establish high-density genetic linkage maps [67] and genome-wide SNP array [68,69,70]. In this study, we used a high-throughput transcript next-generation sequencing platform to screen candidate SNP markers for low-temperature tolerance traits from the database. The results of the Sequenom MassARRAY’s analysis of different cold-tolerant groups of Taiwan tilapia revealed that in addition to polymorphic identification applications, this fish also possesses the ability to withstand low temperatures significantly.

Among 37 SNP markers, we observed differences in low-temperature tolerance among the different genotypes of CL5212_1 (*clathrin* gene). Clathrin translated from the *clathrin* gene contains heavy and light chains. The mesh-like coat formed by polymerization can capture and transport molecules during receptor-mediated endocytosis and organelle biosynthesis to the cell membrane, thus playing a key role in cellular physiology [71]. There are high proportions of A/A and A/d genotypes in cold-tolerant (CT) strain, while the genotypes of non-cold-tolerant (CS) strain is predominantly d/d, as the SNP locus is located in the UTR of the *clathrin* gene. Although there was no sequence change involving the amino acids of the protein, we unexpectedly observed after further prediction of miRNAs and target gene sequence that the A base of the 3′ end UTR of the *clathrin* gene in the cold-sensitive (CS) strain was missing (genotype is d/d), as were the miRNA regulatory sites nearby. It is speculated that miRNAs might negatively regulate the expression of the gene and decrease protein translation, thus leading to the heredity of the phenotype of the cold-tolerant trait in fish. In the future, we will continue to explore the molecular mechanisms underlying this hypothesis.

Figure 7 presents the consistency of the two quantitative RNA*seq* and qPCR results. Among them, under cold acclimation, the RNA*seq* and qPCR data of the CL5212_1 gene in the brain, liver, muscle, and gill tissues were all up-regulated, although we did identify genetic variation (qualitative) in the SNP marker from the CL5212_1 gene. Cold tolerance is related; however, it remains unknown if its gene expression (quantitative) affects the cold tolerance of fish, and determining this will require further follow-up tests.

Additionally, Koštál et al. [72] determined that arginine can form supramolecular aggregates, thus suggesting that in addition to partially binding to unfolded proteins, it can also inhibit its aggregation under freeze dehydration to stimulate fruit fly larvae with high freeze tolerance. Fan et al. [73] studied arginine kinase (a heat shock protein) based on the genomics and proteomics of *L. vannamei* under cold stress. Proteins and histones were identified as positive regulators. Abdel-Ghany et al. [74] observed that cold shock to Nile tilapia can enhance cold tolerance by reducing the changes in systemic saturated fatty acids and increasing the lipid metabolism of n-6 and n-3 unsaturated fatty acids. Additionally, Li et al. [75] determined that in tilapia diets supplemented with arginine, the regulation of lipid metabolism-related genes can reduce the mechanism of liver fat deposition and fatty acid composition induced by a high-fat diet [76]. From the transcript sequencing results of this study, it was observed that the arginine kinase gene of the cold-tolerant strain possessed high expression levels.

In summary, this study conclusively demonstrated that after long-term collection from different source groups and the establishment of multi-generation seed stocks through family pairing genetic management flags and trait recording data tracking systems, the extremes of genetic variation can be screened from important economic traits of farmed fish in ethnic group individuals [26,77]. The combination of high-throughput transcriptome sequencing and assembly, database comparison and functional annotation, gene differential expression, and molecular marker genotype analysis was quite effective in exploring the theory of key regulatory pathways in stress physiology, the development of molecular marker-assisted breeding platforms and their potential applications [78,79]. In the future, in addition to further research examining the effects of aquatic diet additives on the cold resistance traits of aquatic products, artificial intelligence, and nutrigenomics research strategies such as the AI-assisted adversity abnormal behavior identification system [80] will be used. The recently developed whole genomic SNP array technology is also crucial for the follow-up study of more Taiwan tilapia resistance traits and genome-wide association studies, to improve the research and development of precise nutrition breeding regulation mechanisms.

## 5. Conclusions

In conclusion, the results of this study lay the foundation for identifying genetic markers associated with cold tolerance and for developing molecular markers based on simple sequence repeats and single nucleotide polymorphisms. These can be used in tilapia breeder-assisted selection breeding plans to increase the temperature adaptation range and productivity of tilapia. Tilapia breeding develops tilapia strains possessing improved cold tolerance through marker-assisted selection. This study proposes that multiple sets of novel polymorphic SSR and SNP markers derived from RNA*seq* can be applied to the establishment, monitoring, and management of the genetic resource diversity of various commercial populations of Taiwan tilapia in the market and can also be used for providing the basis for family selection and breeding genetics and to further explore the gene regulation mechanisms and precision breeding of cold-tolerant strains of additional aquaculture species.

## Figures and Tables

**Figure 1 animals-11-03538-f001:**
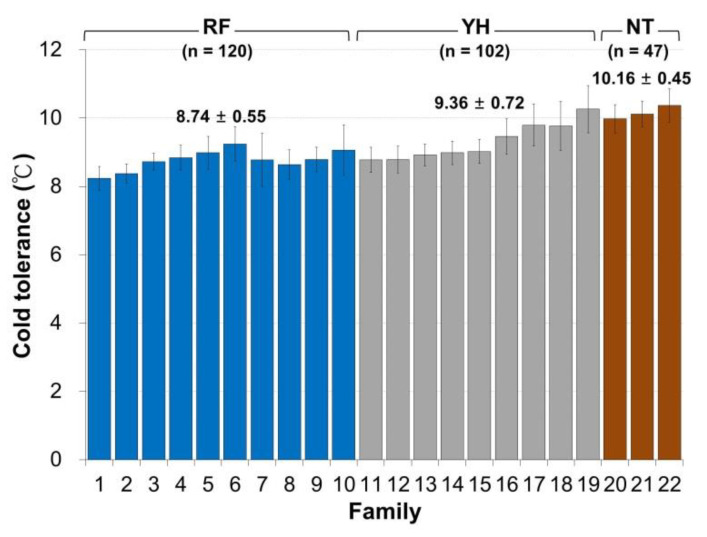
The phenotypic distribution of the measured values under cold conditions based on the cold-tolerance temperature of 22 in the Taiwan tilapia family RF (blue), YH (grey) and NT (brown) populations. The top 3 families (Family 1, 2, and 8) and the last 3 families (Family 19, 21, and 22) were sorted according to cold-tolerance temperature and defined as cold-tolerance (CT) and cold-sensitivity (CS), respectively.

**Figure 2 animals-11-03538-f002:**
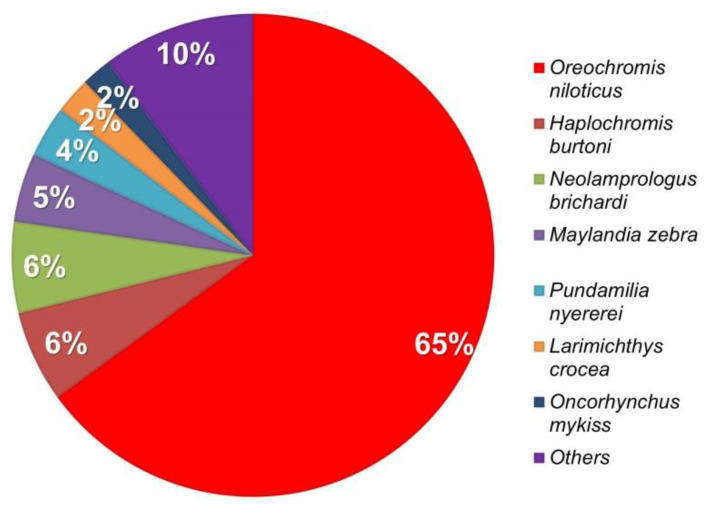
Distribution of annotated species. The distribution of annotated species is characterized statistically using NR annotation. *Oreochromis niloticus*: 64.99%; *Neolamprologus brichardi*: 6.11%; *Haplochromis burtoni*: 6.16%; *Maylandia zebra*: 4.6%; other: 18.14%.

**Figure 3 animals-11-03538-f003:**
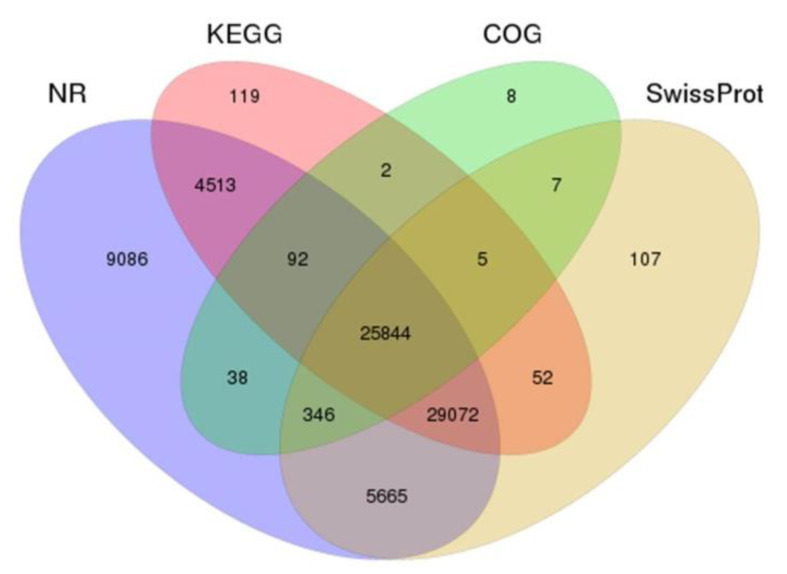
Annotation of assembled Taiwan tilapia unigenes. The unigenes were annotated according to different protein databases that included NR, KEGG, COG and Swiss-Prot and are presented in a Venn diagram. NR: Unigenes with NCBI non-redundant protein. KEGG: Kyoto Encyclopedia of Genes and Genomes. COG: Clusters of Orthologous Groups. Swiss-Prot: A curated protein sequence database which strives to provide a high level of annotation.

**Figure 4 animals-11-03538-f004:**
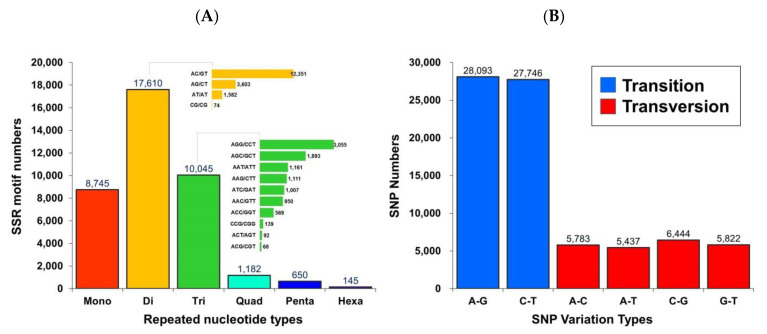
Identification of simple sequence repeats (SSRs) and single nucleotide polymorphism (SNPs) quantity statistics in the transcriptome of Taiwan tilapia. (**A**) The histogram presents the distribution of the total number of SSRs within different classes (repeated nucleotide types). The *X*-axis is the specific repeated nucleotide types, which were defined as mone for one-, di for two-, tri for three-, quad for four-, penta for five-, and hexa for six-nucleotides and the *Y*-axis indicates the SSR motif numbers. (**B**) The histogram presents the distribution of the total number of SNPs within different classes (transition and transversion). The *X*-axis is the SNP variation type, and the *Y*-axis indicates the SNP numbers.

**Figure 5 animals-11-03538-f005:**
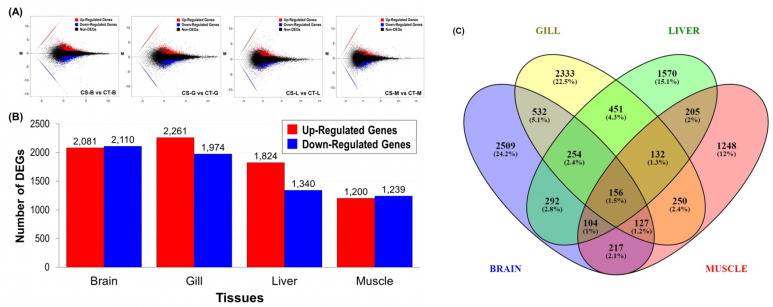
Statistics for the differentially expressed genes (DEGs). (**A**) MA plot of the DEGs. The *X*-axis represents value A (log_2_ transformed mean expression level). The *Y*-axis represents value M (log_2_ transformed fold change). Red, blue, and black points represent up-, down- and non-regulated DEGs, respectively. “CS-B, CS-G, CS-L, and CS-M” were the controls and “CT-B, CT-G, CT-L, and CT-M” were experimental groups in the “CS-vs-CT” paired comparison. (**B**) The numbers of differentially expressed genes (DEGs) between the two comparison groups in brain, gill, liver, and muscle tissues were determined. All DEGs were determined based on the results from the statistical analysis (FDR < 0.05). The *X*-axis represents the comparison of samples. The *Y*-axis represents the DEG numbers. The red color represents up regulated DEGs, and the blue color represents down-regulated DEGs. (**C**) Venn diagram of the expressed unigenes in brain, gill, liver, and muscle tissues. A total of 10,380 unigenes were expressed, and of these, 156 genes were commonly expressed in all the four tissues. Unigenes exhibiting |log_2_ fold change| ≧ 1 were considered to be differentially expressed in each tissue.

**Figure 6 animals-11-03538-f006:**
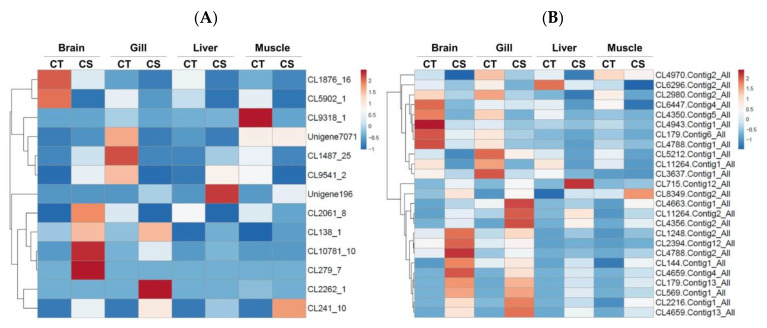
Expression analysis of Taiwan tilapia transcripts containing (**A**) microsatellites (SSRs) and (**B**) single nucleotide polymorphisms (SNPs) loci from brain, gill, liver, and muscle tissues between the cold-tolerance (CT) and cold-sensitivity (CS) groups based on their relative FPKM values. The transcripts were hierarchically clustered based on correlation distance and average linkage method. Blue indicates the lowest level of expression, white indicates an intermediate level of expression, and red indicates the highest level of expression. FPKM, fragments per kilobase of transcript per million mapped reads.

**Figure 7 animals-11-03538-f007:**
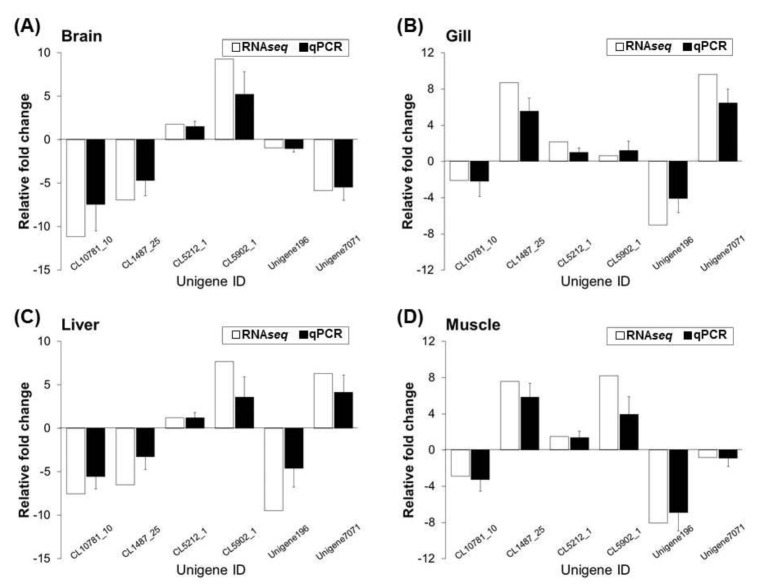
Validation of the FPKM value for the CT and CS of comparative RNA*seq* transcriptome data in brain (**A**), gill (**B**), liver (**C**), and muscle (**D**) tissues using real-time qPCR analysis for relative gene expression. The mRNA expression levels of six unigenes exhibiting different expressions in our RNA*seq* results were confirmed by qPCR analysis-based comparisons of the CT and CS (control) groups. The data are expressed as log fold changes and are represented as standard deviations of the means from three biological replicates.

**Figure 8 animals-11-03538-f008:**
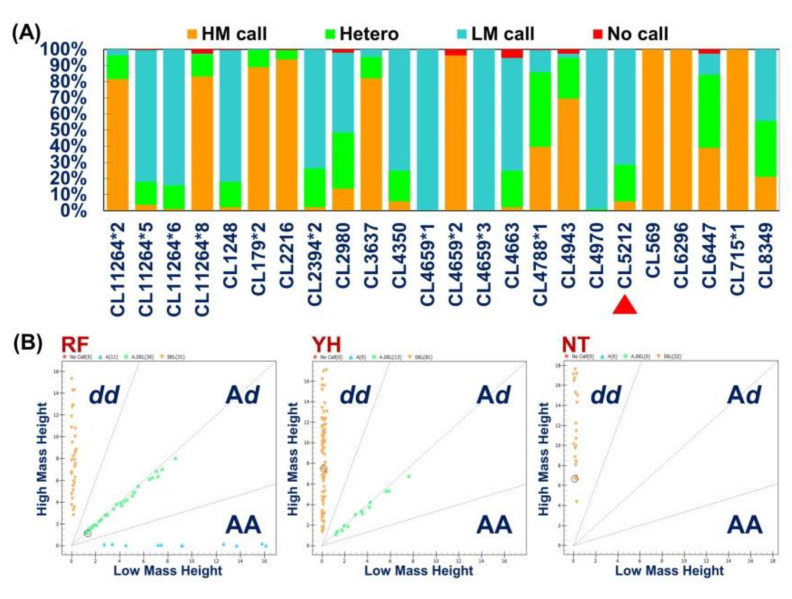
Analysis of SNP markers. (**A**) The frequency of polymorphic and monomorphic SNPs genotyping results derived from 37 genes in Rui-Fang (RF), YiHua (YH), and NTOU (NT) Taiwan tilapia populations. HM call: High mass allele genotype calling; Hetero: Heterozygous genotype calling; LM call: Low mass allele genotype calling; No Calls: Failed for genotyping. The candidate gene marker (CL5212) related to cold-tolerance indicated by red triangles arrow. (**B**) AutoCluster analysis. The call cluster plot displays the single-nucleotide polymorphisms (SNPs) genotype data of 37 targeted unigenes assayed using the Sequenom MassARRAY^®^ platform from 72, 96, and 22 samples in the RF, YH, and NT populations, respectively. One spot corresponds to one sample. Homozygous samples possessing genotypes AA and BB are orange and blue, respectively. Heterozygous samples exhibiting genotype AB are shown in green; samples with missing genotypes are red.

**Figure 9 animals-11-03538-f009:**
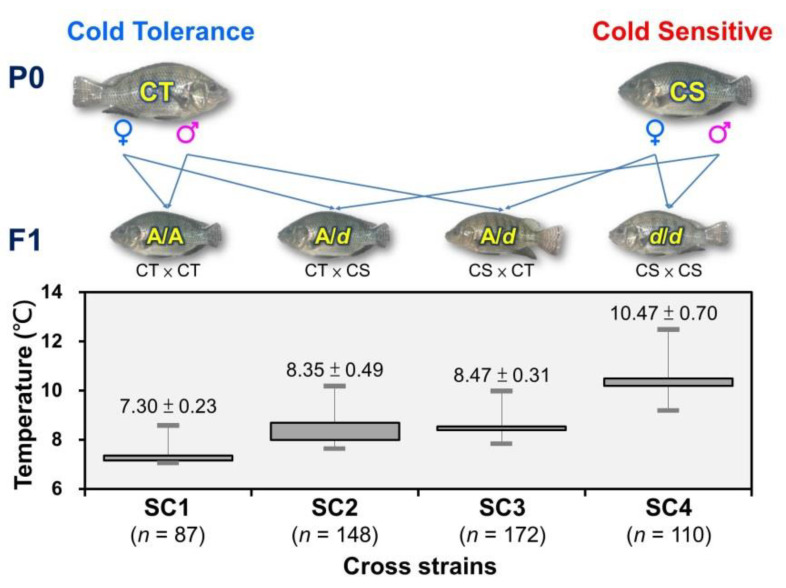
Illustration of the SNP marker-assisted crossbreeding scheme used to obtain the four reciprocal crosses (SC1 [CT × CT], SC2 [CT × CS], SC3 [CS × CT], and SC4 [CS × CS] families [marked as dam × sire]) of CT and CS strains in Taiwan tilapia that were compared for their cold-tolerance in this study. The means (±SD) of the lowest temperature tolerance of the progeny of each hybrid family is presented as a box-and-whisker diagram, and significant differences (*p* < 0.05) between crosses are marked with different letters.

**Table 1 animals-11-03538-t001:** Statistical summary of sequencing reads after filtering in Taiwan tilapia (*Oreochromis* spp.).

Sample ^1^	Clean Reads ^2^	Clean Bases ^3^	Q20 (%) ^4^	Q30 (%) ^5^	GC (%) ^6^	RL (bp) ^7^
CT-B	29,473,122	4,420,968,300	98.88	96.28	49.36	150
CT-G	29,944,552	4,491,682,800	98.69	95.63	51.22	150
CT-L	29,536,538	4,430,480,700	98.76	95.83	50.54	150
CT-M	29,268,400	4,390,260,000	98.9	96.27	52.25	150
CS-B	29,547,582	4,432,137,300	98.72	95.85	49.57	150
CS-G	28,697,764	4,304,664,600	98.8	96.04	49.92	150
CS-L	29,117,880	4,367,682,000	99.05	96.72	50.07	150
CS-M	29,179,716	4,376,957,400	98.81	96.02	52.11	150

^1^ CT-B, brain tissue of the cold-tolerance group; CT-G, gill tissue of the cold-tolerance group; CT-L, liver tissue of the cold-tolerance group; CT-M, muscle tissue of the cold-tolerance group; CS-B, brain tissue of the cold-sensitive group; CS-G, gill tissue of the cold-sensitive group; CS-L, liver tissue of the cold-sensitive group; CS-M, muscle tissue of the cold-sensitive group. ^2^ Total number of paired-end reads after filtering the sequencing reads which containing low-quality, adaptor-polluted and high content of unknown base (N) reads. ^3^ Total clean bases (nt) = Total clean reads × Read length. ^4^ The rate of bases which quality is greater than 20. ^5^ The rate of bases which quality is greater than 30. ^6^ The percentage of G and C bases in all unigenes. ^7^ RL, read length.

**Table 2 animals-11-03538-t002:** Functional annotation information of the Taiwan tilapia transcriptome dataset.

Transcriptome Dataset	Unigene Number	Percentage (%)
NR ^1^	74,656	58.26
NT ^2^	97,575	76.14
Swiss-Prot ^3^	61,098	47.68
COG ^4^	26,342	20.56
CO ^5^	7154	5.58
KEGG ^6^	59,699	46.59
Overall (total annotation) ^7^	100,108	78.12
Total	128,147	100

^1^ NR: Unigenes with NCBI non-redundant protein. ^2^ NT: Nucleotide Database. ^3^ Swiss-Prot: A curated protein sequence database which strives to provide a high level of annotation. ^4^ COG: Clusters of Orthologous Groups. ^5^ CO: Gene Ontology. ^6^ KEGG: Kyoto Encyclopedia of Genes and Genomes. ^7^ The number of unigenes which were annotated in at least one functional database.

**Table 3 animals-11-03538-t003:** Overall information of simple sequence repeat (SSR) marker genes in Taiwan tilapia transcriptome database.

No.	Unigene ID	SSR	Length	Position	LG ^1^	Location	Gene Annotation ^2^
1	CL138_1	(A)_n_	2492	1902	LG7	3′-UTR	CTD small phosphatase-like protein 2-A
2	CL1487_25	(GT)_n_	8664	1862	LG16	5′-UTR	Nuclear pore complex protein Nup98-Nup96 isoform X6
3	CL1876_16	(TTC)_n_	7751	7251	LG4	3′-UTR	Myosin-10 isoform X3
4	CL5902_1	(G)_n_	2572	569	LG12	3′-UTR	Ubiquitin-conjugating enzyme E2 G1
5	CL9541_2	(CAG)_n_	5825	2164	LG17	Exon	AT-rich interactive domain-containing protein 2
6	CL10781_10	(A)_n_	9102	8403	LG8	3′-UTR	Fatty acid synthase isoform X1
7	CL279_7	(GCA)_n_	4772	2266	LG23	Exon	R3H domain-containing protein 1
8	CL2262_1	(T)_n_	3105	3083	LG2	3′-UTR	TNFAIP3-interacting protein 1 isoform X1
9	Unigene7071	(GT)_n_	2729	1722	LG14	Intron	Protein IWS1 homolog isoform X3
10	Unigene196	(CA)_n_	5125	5073	LG14	3′-UTR	Serine/threonine-protein kinase SIK3 isoform X3
11	CL2061_8	(TC)_n_	1393	102	LG7	5′-UTR	Glucose-6-phosphate isomerase-like
12	CL9318_1	(ATT)_n_	706	530	LG12	3′-UTR	Bifunctional methylenetetrahydrofolate dehydrogenase/cyclohydrolase, mitochondrial-like
13	CL241_10	(GAG)_n_	5371	3079	LG8	Exon	Rho GTPase-activating protein 17 isoform X1

^1^ Linkage group. ^2^ Assembled unigenes were aligned against the NCBI non-redundant nucleotide sequence database (Nt) by BLASTn with an E-value cut off of 10−5. Then they were searched in public databases including NR, COG, GO, and KEGG through BLASTx under the same criterion as BLASTn.

**Table 4 animals-11-03538-t004:** Overall information of single nucleotide polymorphism (SNP) marker genes in Taiwan tilapia transcriptome database.

SNP	Unigene ID	Length	LG ^1^	Allele	Location	Change ^2^	Gene Annotation ^3^
1	CL4970	3236	2	T/C	UTR	-	pre-mRNA-splicing factor RBM22
2	CL6296_2	3447	23	d/A	UTR	-	lipoamide acyltransferase component of branched-chain alpha-keto acid dehydrogenase complex, mitochondrial
3	CL2980_2	1540	3	G/C	Exon	P/P	NADH dehydrogenase
4	CL6447_4	3573	ND	C/T	Intron	-	WD repeat domain phosphoinositide-interacting protein 2 isoform X1
5	CL4350_5	3286	23	G/A	UTR	-	protein Jade-3
6	CL4943_1	2166	2	A/G	UTR	-	NEDD4 family-interacting protein 1 isoform X2
7	CL179_6	5803	5	G/A	Exon	V/I	period circadian protein homolog 3-like isoform X4
8	CL4788_1	1945	4	d/G	UTR	-	hsc70-interacting protein isoform X2
9	CL4788_1	1945	4	d/A	UTR	-	
10	CL5212_1	1774	2	d/A	UTR	-	clathrin light chain B-like isoform X4
11	CL11264_1	2075	2	G/A	UTR	-	cyclin-G1-like
12	CL11264_1	2075	2	C/T	UTR	-	
13	CL11264_1	2075	2	C/A	UTR	-	
14	CL11264_1	2075	2	T/C	UTR	-	
15	CL11264_1	2075	2	G/A	Exon	I/V	
16	CL3637_1	10,003	23	T/A	UTR	-	protein PRRC2C
17	CL715_12	5227	13	d/A	Exon	S/K	adipocyte plasma membrane-associated protein isoform X2
18	CL715_12	5227	13	d/C	UTR	-	
19	CL715_12	5227	13	d/C	UTR	-	
20	CL715_12	5227	13	d/G	UTR	-	
21	CL8349_2	3478	7	T/C	UTR	-	ATP-dependent Clp protease ATP-binding subunit clpX-like, mitochondrial isoform X1
22	CL4663_1	3831	ND	A/T	UTR	-	bone morphogenetic protein receptor type-1A
23	CL11264_2	2074	2	A/G	UTR	-	cyclin-G1-like
24	CL11264_2	2074	2	C/T	UTR	-	
25	CL11264_2	2074	2	G/T	UTR	-	
26	CL4356_2	1809	ND	d/G	UTR	-	bone morphogenetic protein receptor type-1A
27	CL1248_2	3831	2	C/T	Intron	-	dynactin subunit 4 isoform X2
28	CL2394_12	8460	13	G/A	Exon	R/H	spectrin beta chain, non-erythrocytic 1
29	CL2394_12	8460	13	C/T	Exon	V/I	
30	CL4788_2	2157	4	A/G	Exon	A/A	hsc70-interacting protein isoform X2
31	CL144_1	1772	13	d/G	UTR	-	E3 ubiquitin-protein ligase UBR2 isoform X3
32	CL4659_4	3929	13	d/T	Exon	F/F	echinoderm microtubule-associated protein-like 4 isoform X3
33	CL179_13	1906	5	d/T	UTR	-	period circadian protein homolog 3-like isoform X4
34	CL569_1	3529	13	d/A	Exon	P/P	cullin-9 isoform X1
35	CL2216_1	1929	8	T/C	Exon	S/T	dynamin-binding protein isoform X3
36	CL4659_13	4077	13	d/T	Exon	D/D	echinoderm microtubule-associated protein-like 4 isoform X3
37	CL4659_13	4077	13	d/A	Exon	T/N	

^1^ Linkage group. ^2^ Amino acid change. ^3^ Assembled unigenes were aligned against the NCBI non-redundant nucleotide sequence database (NT) by BLASTn with an E-value cut off of 10^−5^. Then they were searched in public databases including NR, COG, GO, and KEGG through BLASTx under the same criterion as BLASTn.

**Table 5 animals-11-03538-t005:** Genomic SSR markers used for genetic variation study in RF, YH, and NT populations of Taiwan tilapia.

Locus ^1^	*Ho*			*He*			*PIC*			*F_IS_*			*p* Value ^2^
	RF	YH	NT	RF	YH	NT	RF	YH	NT	RF	YH	NT	
UNH916	0.722	0.480	0.532	0.775	0.485	0.395	0.769	0.483	0.390	0.061	0.004	−0.362	0.345
UNH999	0.778	0.686	0.447	0.767	0.671	0.395	0.761	0.668	0.390	−0.022	−0.028	−0.144	0.103
CL1487_25	0.472	0.726	0.000	0.624	0.676	0.000	0.620	0.672	0.000	0.239	−0.079	—	0.537
CL1876_16	0.181	0.000	0.000	0.289	0.000	0.000	0.287	0.000	0.000	0.371	—	—	0.004 **
CL5902_1	0.903	0.441	0.000	0.753	0.658	0.000	0.747	0.655	0.000	−0.208	0.326	—	0.456
CL9541_2	0.472	0.598	0.447	0.523	0.472	0.351	0.520	0.470	0.347	0.091	−0.273	−0.288	0.217
CL10781_10	0.278	0.559	0.489	0.412	0.592	0.599	0.410	0.589	0.593	0.322	0.051	0.175	0.178
CL279_7	0.000	0.000	0.000	0.130	0.000	0.000	0.129	0.000	0.000	1.000	—	—	0.017 *
CL2262_1	0.722	0.784	0.894	0.798	0.728	0.678	0.793	0.725	0.671	0.089	−0.082	−0.332	0.257
Unigene7071	0.694	0.196	0.575	0.749	0.201	0.431	0.744	0.200	0.427	0.066	0.021	−0.346	0.011 *
Unigene196	0.750	0.686	0.553	0.776	0.710	0.454	0.771	0.706	0.449	0.027	0.028	−0.232	0.098
CL9318_1	0.667	0.284	0.000	0.759	0.283	0.000	0.754	0.282	0.000	0.116	−0.010	—	0.010 *
CL241_10	0.319	0.382	0.575	0.383	0.367	0.505	0.381	0.365	0.499	0.161	−0.048	−0.151	0.255
Mean	0.535	0.448	0.347	0.595	0.449	0.293	0.591	0.447	0.290	0.178	−0.008	−0.210	
SD	0.272	0.264	0.306	0.225	0.260	0.256	0.223	0.259	0.253	0.277	0.135	0.165	

^1^ Locus symbol is derived from reference or abbreviated according to the unigene ID form transcriptome database. ^2^ The *p* value is mean level of significance. Data sets that are significant at different levels: * *p* < 0.05, ** *p* < 0.01.

**Table 6 animals-11-03538-t006:** The number of genotype, the frequency of homozygous and heterozygous, and the overall data of cold-tolerance related tests were analyzed by using RNA*seq* single nucleotide polymorphism (SNP) marker loci in RF, YH and NT Taiwan tilapia populations.

Unigene	SNP	High Mass Allele Calls			Heterozygous Calls			Low Mass Allele Calls			Homozygous *freq*			Heterozygous *freq*			HW ChiSquare	HW *p*-Value	Trait *p*-Value
		RF	YH	NT	RF	YH	NT	RF	YH	NT	RF	YH	NT	RF	YH	NT			
CL4970	C/T	0	0	0	0	2	0	72	94	22	1.000	0.979	1.000	0.000	0.021	0.000	0.01	0.940	0.879
CL6296_2	A/*d*	0	0	0	0	0	0	72	96	22	1.000	1.000	1.000	0.000	0.000	0.000	-	-	-
CL2980_2	C/G	13	13	0	31	35	0	27	45	22	0.563	0.624	1.000	0.437	0.376	0.000	6.08	0.010	0.148
CL6447_4	C/T	25	49	0	28	36	22	15	10	0	0.588	0.621	0.000	0.412	0.379	1.000	0.00	1.000	0.056
CL4350_5	A/G	11	0	0	23	0	13	38	96	9	0.681	1.000	0.409	0.319	0.000	0.591	13.60	0.000	0.589
CL4943_1	A/G	37	73	22	28	20	0	2	3	0	0.582	0.792	1.000	0.418	0.208	0.000	0.06	0.800	0.919
CL179_6	G/A	58	89	22	13	7	0	1	0	0	0.819	0.927	1.000	0.181	0.073	0.000	0.23	0.630	0.591
CL4788_1_1	G/*d*	16	10	0	29	37	22	27	48	0	0.597	0.611	0.000	0.403	0.389	1.000	0.00	0.980	0.094
CL4788_1_2	A/*d*	0	0	0	0	0	0	72	96	22	1.000	1.000	1.000	0.000	0.000	0.000	-	-	-
CL5212_1	A/*d*	11	0	0	30	13	0	31	83	22	0.583	0.865	1.000	0.417	0.135	0.000	7.75	0.010	0.001 **
CL11264_1_1	G/A	38	95	22	28	1	0	6	0	0	0.611	0.990	1.000	0.389	0.010	0.000	8.15	0.000	0.378
CL11264_1_2	C/T	38	95	22	27	1	0	7	0	0	0.625	0.990	1.000	0.375	0.010	0.000	11.92	0.000	0.602
CL11264_1_3	C/A	37	95	22	27	1	0	7	0	0	0.620	0.990	1.000	0.380	0.010	0.000	11.81	0.000	0.599
CL11264_1_4	T/C	1	0	0	25	1	0	46	95	22	0.653	0.990	1.000	0.347	0.010	0.000	0.00	0.970	0.650
CL11264_1_5	G/A	7	0	0	26	1	0	38	95	22	0.634	0.990	1.000	0.366	0.010	0.000	12.91	0.000	0.490
CL3637_1	T/A	38	96	22	25	0	0	9	0	0	0.653	1.000	1.000	0.347	0.000	0.000	22.54	0.000	0.473
CL715_12_1	A/*d*	0	0	0	0	0	0	72	96	22	1.000	1.000	1.000	0.000	0.000	0.000	-	-	-
CL715_12_2	C/*d*	28	7	0	35	43	22	9	46	0	0.514	0.552	0.000	0.486	0.448	1.000	242.63	0.000	0.099
CL715_12_3	C/*d*	9	0	0	24	0	22	38	96	0	0.662	1.000	0.000	0.338	0.000	1.000	200.20	0.000	0.488
CL715_12_4	G/*d*	0	0	0	0	0	0	67	96	22	1.000	1.000	1.000	0.000	0.000	0.000	-	-	-
CL8349_2	C/T	0	28	12	6	50	10	66	18	0	0.917	0.479	0.545	0.083	0.521	0.455	13.43	0.000	0.366
CL4663_1	T/A	4	0	0	21	0	22	38	95	0	0.667	1.000	0.000	0.333	0.000	1.000	0.06	0.810	0.769
CL11264_2_1	G/A	2	0	0	27	1	0	43	95	22	0.625	0.990	1.000	0.375	0.010	0.000	0.38	0.540	0.650
CL11264_2_2	C/T	46	95	22	25	1	0	1	0	0	0.653	0.990	1.000	0.347	0.010	0.000	0.00	0.970	0.650
CL11264_2_3	G/T	41	95	22	25	1	0	1	0	0	0.627	0.990	1.000	0.373	0.010	0.000	0.00	0.950	0.650
CL4356_2	G/*d*	0	0	0	0	0	0	72	96	22	1.000	1.000	1.000	0.000	0.000	0.000	-	-	-
CL1248_2	T/C	4	0	0	27	3	0	40	93	22	0.620	0.969	1.000	0.380	0.031	0.000	2.83	0.090	0.581
CL2394_12_1	T/C	4	0	0	24	9	22	44	87	0	0.667	0.906	0.000	0.333	0.094	1.000	0.41	0.520	0.888
CL2394_12_2	A/G	4	0	0	24	0	22	44	96	0	0.667	1.000	0.000	0.333	0.000	1.000	0.01	0.920	0.888
CL4788_2	G/A	23	80	14	35	15	8	14	1	0	0.514	0.844	0.636	0.486	0.156	0.364	3.85	0.050	0.104
CL144_1	G/*d*	0	0	0	0	0	0	70	96	22	1.000	1.000	1.000	0.000	0.000	0.000	-	-	-
CL4659_4	T/*d*	4	0	0	23	0	22	44	96	0	0.676	1.000	0.000	0.324	0.000	1.000	199.71	0.000	0.926
CL179_13	T/*d*	0	0	0	0	0	0	72	96	22	1.000	1.000	1.000	0.000	0.000	0.000	-	-	-
CL569_1	A/*d*	0	0	0	0	0	0	72	96	22	1.000	1.000	1.000	0.000	0.000	0.000	-	-	-
CL2216_1	T/C	70	96	12	1	0	10	0	0	0	0.986	1.000	0.545	0.014	0.000	0.455	0.17	0.680	0.478
CL4659_13_1	T/*d*	0	0	0	0	0	0	72	96	22	1.000	1.000	1.000	0.000	0.000	0.000	-	-	-
CL4659_13_2	A/*d*	0	0	0	0	0	0	66	95	22	1.000	1.000	1.000	0.000	0.000	0.000	-	-	-

The *p* value is mean level of significance. Data sets that are significant at different levels: ** *p* < 0.01.

## Data Availability

All relevant data are available within the article and its Appendix A.

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
