# Peer review of "Identification of Genes Related to Cold Tolerance and Novel Genetic Markers for Molecular Breeding in Taiwan Tilapia (Oreochromis spp.) via Transcriptome Analysis"

_animals, 2021, doi:10.3390/ani11123538_

Round 1

Reviewer 1 Report

Authors have taken a praiseworthy attempt to understand cold tolerance in fish. However there are several things need to be taken care of.

Major comment

  1. The introduction is too big and out of scope of this paper
  2. its better to discuss some of the candidate pathway more logically
  3. needs serious English editing

Minor comments

  1. Better to use art least two housekeeping gene as internal control instead of one
  2. History or Taiwan tilapia should be properly cited.

Author Response

請參閱附件。

Reviewer 2 Report

I believe that it is a very interesting work and that it achieves important results in relation to the tolerance or sensitivity to temperature stress in a fish that is a resource widely used in world aquaculture. The experimental design from the point of view of genetic improvement and through the use of transcriptomics for the detection of genomic markers involved in the regulation of this important function has been very complete and adequate with the objectives set. I have only suggested small changes in the English edition or other small details in the text. 
